# Decomposition of Water Level Time Series of a Tidal River into Tide, Wave and Rainfall-Runoff Components

**Myungjin Lee** [1] , **Younghoon You** [1], **Soojun Kim** [1,*], **Kyung Tak Kim** [2] **and Hung Soo Kim** [1]

[1]  Department of Civil Engineering, Inha University, Incheon 22212, Korea; lmj3544@naver.com (M.L.); dudgns5971@naver.com (Y.Y.); sookim@inha.ac.kr (H.S.K.)
[2]  Department of Land, Water and Environment Research, Korea Institute of Civil Engineering and Building Technology (KICT), Goyang-Si 10223, Korea; kinha2117@gmail.com
*   Correspondence: sk325@inha.ac.kr; Tel.: +82-10-5404-3538

**Abstract:** The water-level time series of a tidal river is influenced by various factors and has a complex structure, which limits its use as hydrological forecast data. This study proposes a methodology for decomposing the water-level time series of a tidal river into various components that influence the water level. To this end, the tide, wave, rainfall-induced runoff and noise components were selected as the main components that affect the water-level time series. The tide component and the wave component were first separated through wavelet analysis and curve fitting and then they were removed from the water-level data. A high-pass filter was then applied to the resulting time series to separate the rainfall-induced runoff component and the noise component. These methods made it possible to determine the rate of influence that each component has on the water level of a tidal river. The results could be used as a basis for calibrating a rainfall-runoff model and issuing flood forecasts and warnings for a tidal river.

**Keywords:** tidal river; tide; wave; rainfall-runoff; wavelet analysis; entropy

## 1. Introduction

A tidal river is affected by the tides due to the influence of the sun and the moon and its water level is characterized by periodical rising and falling [1]. Thus, flood damage can be greater at high tide due to the backwater effect. However, it is very difficult to measure the flood discharge quantitatively due to the possibility of reverse discharge (i.e., flow from downstream to upstream; minus discharge), which depends on the tide. Researchers have therefore conducted studies to protecting human life and property from natural disasters such as flooding and inundation due to fluctuations in sea level in coastal areas [2–4].

Hidayat et al. constructed artificial neural network (ANN) models using upstream water level data and tide level data to predict the discharge of a tidal river. They obtained good performance for predictions for up to two days in advance [5]. Fan et al. used the Qual2K and HEC-RAS models to assess the water quality of a tidal river in northern Taiwan and found that water quality varies according to the tidal effect [6]. Kim and Kim (2013) studied the spatiotemporal variability characteristics of daily mean tidal residuals on the coasts of Korea. They reported that the variability characteristics on Korea's Yellow Sea coastline are dominated by the influence of wind along the north-south axis rather than changes in atmospheric pressure [7].

To determine tidal residuals accurately, it is important to isolate the tide level time series and determine its characteristics. Liu showed that the energy distribution corresponding to the frequencies

shown in the Fourier spectrum is not always constant over a 10-min period. They showed intermittent increases and decreases according to sea-surface fluctuations [8]. The study noted that this intermittent nature of wave groupings, which characterizes the fluctuations appearing in wind wave recordings, is not shown in the frequency spectrum.

Kang and Lin (2007) conducted a wavelet analysis on three kinds of hydrological data: precipitation, stream flow and well water levels. Although there was no temporal pattern in the case of precipitation, such patterns were present in the stream flow and well water levels [9]. Kang et al. (2013) used Fourier interpretation and wavelet analysis to shed light on tidal residual characteristics in coastal waters. As a result, tidal residual components were sorted into those with short periods of 24 h or less, mid-periods of 1 to 16 days and long periods of 1 month or more. They found that the mid-period components were largely affected by seasonal winds [10].

Research has mainly been conducted on tide levels and waves in coastal areas and there has been a lack of interest in how the water level of a tidal river is influenced by various components [11–13]. Therefore, in this study, we propose a methodology for dividing the water level data of a tidal river into four components: tide, wave, rainfall runoff and noise. The influence of these components was analyzed.

## 2. Methodology

A tidal river has complex time-series characteristics due to the influence of various factors. We employed a methodology to decompose the water level of a tidal river into distinct components (see Figure 1). We judged that the water level data of a tidal river are mainly influenced by the tides, waves, rainfall runoff and noise and we sorted the data in terms of these components. We can use the water-level time series of a river and the sea level time series of the sea in this case, which have similar statistical characteristics because they are close to each other. First, we did a periodicity analysis to separate the periodic tide component and the non-periodic wave component. Next, we removed these components from the water level of the river and then used filtering analysis to separate the rainfall-induced runoff component and the noise component. The study framework is shown in Figure 1.

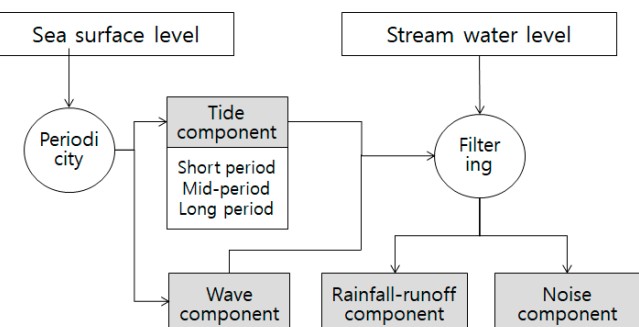

**Figure 1.** Study framework.

### 2.1. Wavelet Analysis

The wavelet transform was first proposed by Haar in the early 1900s [5,14,15]. It is an orthogonal transformation method like the Fourier transform, which has been the most widely used method in the field of signal processing [16]. The wavelet transform can decompose data or a given function into various frequency components and is used to interpret each of these components according to the resolution corresponding to its scale [17–19]. Unlike frequency spectrum analysis, wavelet analysis includes time information. Thus, it can be used to analyze a spectrum's temporal characteristics and facilitates the analysis of transient and irregular wave states. In the case of Fourier transforms, a new basis space is constituted by basic functions like sine, cosine, or exponential functions. In the case of wavelet transforms, however, it is constituted by new basis functions called mother wavelets [20,21].

Therefore, Fourier transforms are localized in only the frequency domain, whereas wavelet transforms are localized in both the frequency and time domains [22]. This is a key factor in accounting for the phenomenon of non-stationary states.

The mother wavelet $\psi(t)$ is mathematically expressed as follows:

$$\psi_{a,b}(t) = \frac{1}{\sqrt{|a|}}\psi\left(\frac{t-b}{a}\right) \quad a,b \in R \tag{1}$$

This equation represents the scaling and translation of the basis function. $a$ is a value that determines the scaling and $b$ is a value that determines how much to shift the function. Equation (1) uses a wavelet function that shifts the mother wavelet by $b$ and changes its scale by $a$ [23]. The width of a wavelet's function becomes narrower at higher frequencies and wider at lower frequencies. The wavelet transform expresses an arbitrary function through the superposition of wavelet basis functions and each function has a different scale level with a resolution corresponding to that level.

### 2.2. Curve Fitting

Given a set of non-stationary data that is realistically obtainable, curve fitting refers to the method of finding a mathematical straight line or curve that can best represent the data points in that set. Curve fitting is used as a core technique for the analysis of experimental data in a wide range of fields, such as science and engineering, statistics and various automation technologies. Its value has risen in tandem with the rapid advances in computer technology since the 1980s.

There are two main types of curve fitting: least-squares regression and interpolation. Least-squares regression is used when the data contain a significant degree of error or scattered data points but the method has a drawback in that the derived curve cannot fit all the data points. Interpolation is used when there is very precise knowledge of the data. Interpolation methods are divided into linear interpolation and polynomial interpolation. We isolated tide-level components that undergo periodic oscillations that are describable by sine and cosine functions by using a polynomial interpolation that consists of sine functions:

$$f(t) = a_1 sin(b_1 \times x + c_1) + a_2 sin(b_2 \times x + c_2) + \cdots + a_n sin(b_n \times x + c_n) \tag{2}$$

### 2.3. Filters

A filter can be used to limit the noise from the frequency range of a desired signal and involves the removal of unwanted portions from the frequency spectrum [24]. Filters are generally divided into four types: low-pass, high-pass, band-pass and band-reject filters. As shown in Figure 2, components with a high frequency can be extracted with a high pass filter and components with a low frequency can be extracted with a low pass filter. The aim in this study was partly to extract the long-period component of the tide level and we used a high-pass filter that passes signals in a band ranging from a certain cutoff frequency to infinity.

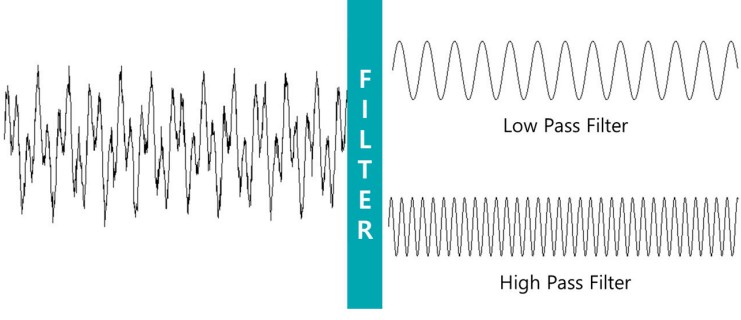

**Figure 2.** Concept of Filtering.

### 2.4. Entropy Theory for Information Measurement

Shannon and Weaver [25,26] defined the marginal entropy, as shown in Equation (3):

$$H(X) = -\sum_{n=1}^{N} p(x_n) \ln p(x_n), \; n = 1, 2, 3, \cdots, N \tag{3}$$

where $p(x_n)$ is the occurrence probability of $x_n$ and $H(X)$ is the marginal entropy that represents the amount of information of $X$.

If a variable $y_m(m = 1, 2, \cdots, N)$ has information related to a variable $x_n$, the uncertainty of $x_n$ may be reduced. Based on the theory, the conditional entropy $H(X|Y)$ in $X$ with the given $Y$ can be estimated [27], as shown in Equation (4):

$$H(X|Y) = -\sum_{n=1}^{N} \sum_{m=1}^{N} p(x_n, y_m) \ln p(x_n|y_m) \tag{4}$$

where $p(x_n, y_m)$ is the joint probability of $X = \{x_n\}$ and $Y = \{y_m\}$ and $p(x_n|y_m)$ is the conditional probability of $X$ with the given $Y$. The information transferred between $X$ and $Y$ is defined in Equation (5):

$$T(X, Y) = H(X) - H(X|Y) \tag{5}$$

The information sent from $X$ to $Y$ is defined as $S(X, Y)$:

$$S(X, Y) = \frac{T(X, Y)}{H(Y)} \tag{6}$$

where $H(Y)$ is the marginal entropy of a single variable $Y$ and $T(X, Y)$ is the trans-information between $X$ and $Y$.

## 3. Application and Results

### 3.1. Study Area and Data Collection

The Ulsan basin was selected as the study area. As shown in Figure 3, the Ulsan water level station is on the river in the area, which has tidal river characteristics. There is also a sea level station at the end of the river near the water level station. The Guyeong water level station is located upstream on a part of the river that is not affected by the tide level. Therefore, it is easy to apply and verify the presented methodology in this area.

We collected Ulsan and Guyeong's stream water-level time-series data and Ulsan's sea-level time-series data in units of time (52,562 h). The collected data ranged from 1 January 2011 to 31 December 2016. The data series from the Guyeong stream water level station exhibited steep rises and declines according to the occurrence of flood events without any tidal effects. The data from the Ulsan stream water level station exhibited typical tidal river characteristics with marked fluctuations at low water levels as well as susceptibility to flood events.

The sea level time series data from the Ulsan sea level station did not show a wide range of fluctuations but there was a mixture of periodic and non-periodic components due to both tidal and wave influences. The standard deviation of the time series data from the Ulsan stream water level station was 5.72, that of the Guyeong stream water level station was 1.99 and that of the Ulsan sea level station was 2.15. This shows that the data from the Ulsan stream water level station has the greatest variability. The time series data from the Ulsan stream water level station and sea level station had a Pearson correlation coefficient of 0.85, root mean squared error (RMSE) of 0.54 m for normalized values ($x - \mu/\sigma$) and an RMSE of 0.39 m at low water levels (<1.5 m). This shows that the variability characteristics are quite similar.

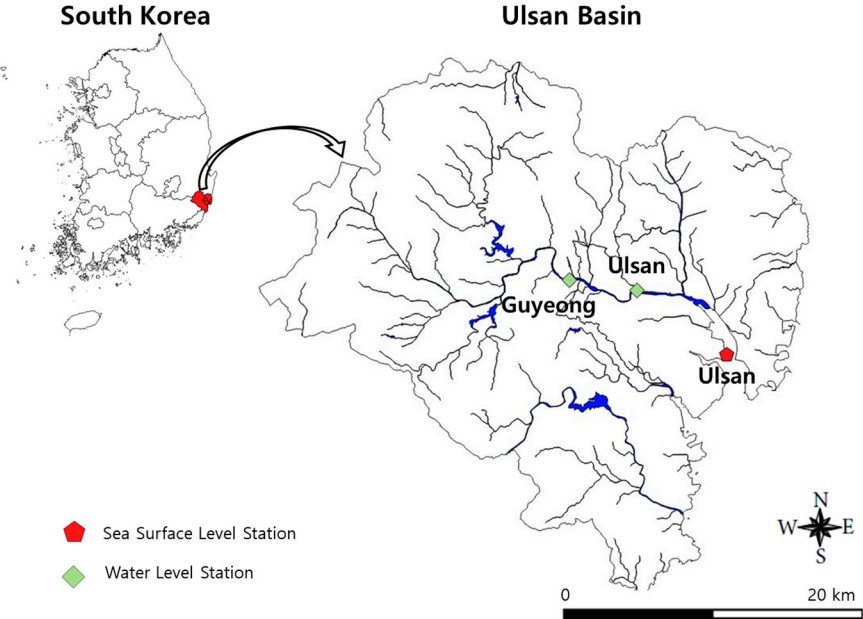

**Figure 3.** Stream water level and sea surface level stations in Ulsan.

*3.2. Separation of the Tide Component and Wave Component*

There is a great deal of uncertainty in hydrological data such as precipitation and water levels, which results in lower predictive accuracy. However, sea level data are generated in part from the movement of the sun and the moon and may thus be regarded as deterministic data. The wave component of the sea level data consists of stochastic data that cause uncertainty in the deterministic nature of the tide level data.

We used wavelet analysis to divide the sea level data into a tide component and a wave component. The tide data consist of short-period components like the semi-diurnal tide, which occurs twice a day; mid-period components like the spring tide or the neap tide, which occurs in 15-day periods; and long-period components that involve longer periods [14,16,17]. Wavelet analysis, hourly data and daily data were used to extract the short-period component and mid-period component. The main idea is that the stream water level is influenced by the sea level and that the sea water level can be decomposed into tide and wave components. Therefore, the water-level time series can also be decomposed partly into tide and wave components.

As shown in Figure 4, the stream water level and sea level have a similar pattern at low water levels. Since there are errors between the two time series, however, we calibrated these errors by normalizing and calibrating the two sets of time-series data as follows:

$$\gamma_t = \theta_t \times \frac{s_1}{s_2} + (\mu_1 - \mu_2) \tag{7}$$

where $\gamma_t$ is the revised sea-level time series, $\theta_t$ is the original sea-level time series, $\mu_1$ and $s_1$ are the mean and standard deviation of Ulsan's stream water level data and $\mu_2$ and $s_2$ are the mean and standard deviation of the sea level data.

We conducted a wavelet analysis to decompose the sea level component in Ulsan's water-level time series into periodic components and a wave component. Due to edge effects, it is possible to use short segments of data such as time series in wavelet analysis but there are limits to extracting long-period components. Thus, we conducted the wavelet analysis separately for hourly data and daily data and the results are shown in Figures 5 and 6. The most important variable component was the 10 to 14-h periodic component for the hourly data and the 13 to 17-day periodic component for the daily data. Therefore, we extracted only the 10 to 14-h short-period tide component and the 13 to 17-day mid-period tide component.

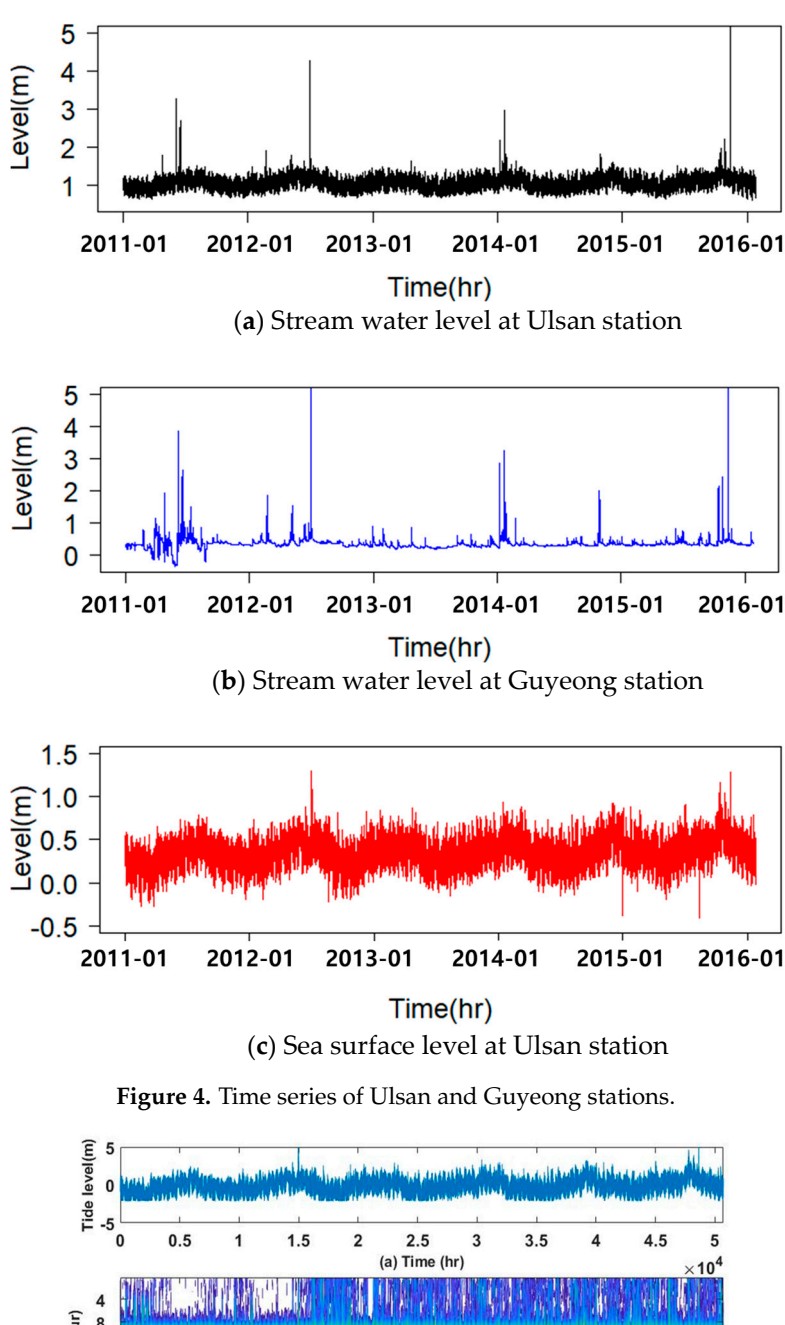

**Figure 4.** Time series of Ulsan and Guyeong stations.

**Figure 5.** Wavelet analysis result of hourly data: (**a**) Ulsan tide level time series, (**b**) Power spectrum of Ulsan tide level, (**c**) 10–14 h scale-average time series. Yellow lines represent meaningful signal of power spectrum. The 24 h component could not be extracted because this component does not satisfy the 95% confidence level.

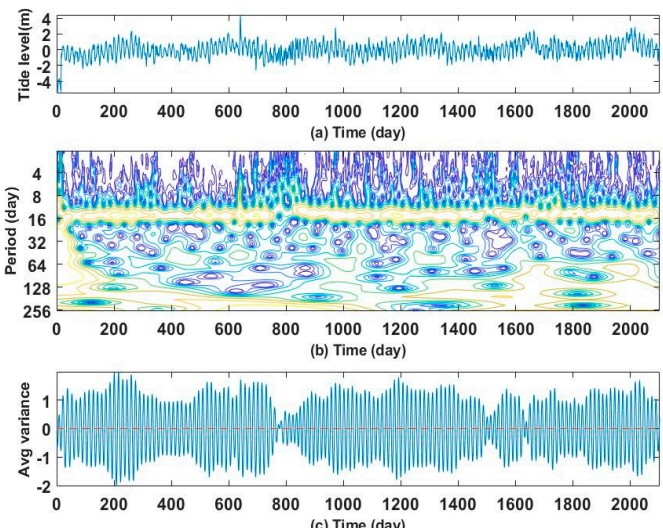

**Figure 6.** Wavelet analysis of daily data: (**a**) Ulsan tide level time series, (**b**) Power spectrum of Ulsan tide level, (**c**) 13–17 day scale-average time series. 13–17 days component was extracted because only these components satisfy the 95% confidence level.

In the tide-level components, the short-period and mid-period components are largely influenced by the moon revolving around the earth. In contrast, the tide level in relation to the long period of approximately one year is influenced by the position of the earth revolving around the sun. Accordingly, even though we can apply wavelet analysis to extract the long-period component, it is feasible only if we obtain a sufficient amount of time-series data. We had access to only about six years of data, however, which imposes a limitation on isolating the long-period component through wavelet analysis. To overcome this limitation, we used a curve-fitting function to express the long-period component as the sum of sine functions:

$$
\begin{aligned}
Long\ Period\ Component &= 1.124 \times sin(0.00005058t - 0.07916) \\
&+ 0.7595 \times sin(0.0000646t + 2.651) + 0.1046 \times sin(0.0007501t - 2.567)
\end{aligned}
\tag{8}
$$

We can then decompose the tide component through wavelet analysis and a fitting function into short-period, mid-period and long-period components. These components can be integrated to represent the integrated tide component. The difference between Ulsan's sea-level time series and the tide component represents the wave component, as shown in Figure 7.

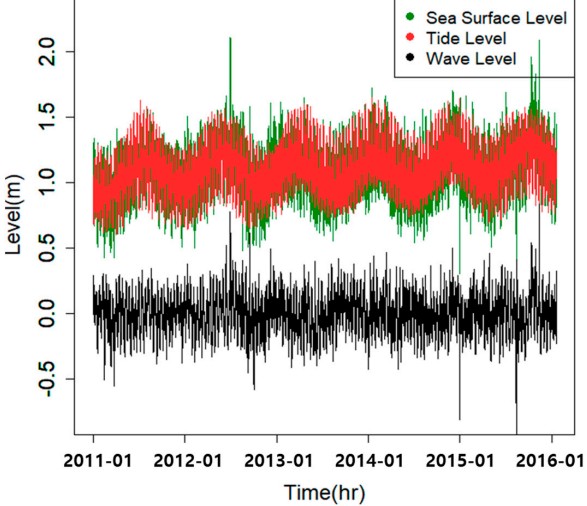

**Figure 7.** Wave component at Ulsan station.

*3.3. Isolation of the Rainfall-Runoff Component*

When the separated tide and wave components are removed, only the rainfall-runoff component and the noise component are left (see Figure 8a). We applied a filter function to separate the runoff component, which occurs in response to non-periodic rainfall events and the noise component, which exhibits random characteristics. We applied a high-pass filter to separate the high-frequency noise component from the relatively low-frequency rainfall-runoff component. The results are shown in Figure 8, with Figure 8b showing the time series for the noise component and Figure 8c showing the rainfall-runoff component.

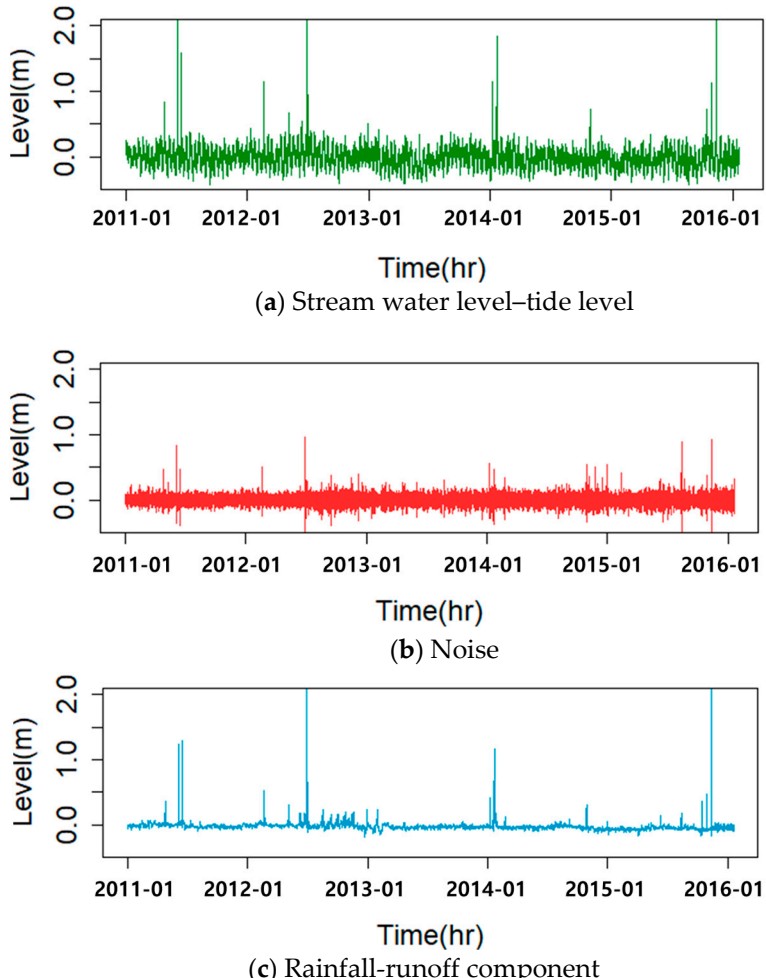

(**a**) Stream water level–tide level

(**b**) Noise

(**c**) Rainfall-runoff component

**Figure 8.** Noise and rainfall-runoff components. High pass filter was applied because the noise component has a high frequency characteristic as shown in Figure 2. Finally, Rainfall-runoff component could be separated by subtracting the value from (**a**) Stream water level–tide level to (**b**) Noise.

## 4. Results and Discussion

In this study, we applied wavelet analysis and a sine fitting function to show that the tide and wave components can be separated from the stream water level data and that the rainfall-runoff and noise components can be separated by using a high-pass filter. We applied each analytical method to the water-level time series obtained from stations in Ulsan and Guyeong and separated four components (tide, wave, rainfall-runoff and noise). The main results are summarized in Figure 9 and the rate at which each component influences the fluctuations in the water-level time series is shown in Table 1.

The ratios in Table 1 are the average of the calculated ratios of each component divided by the stream water level in the entire time series, as in Equation (9).

$$ratio(j) = \frac{1}{n} \sum_{t=1}^{n} \frac{Component(j)_t}{Runoff_t} \tag{9}$$

In the water-level time series (Figure 9a), fluctuations at low water levels were largely influenced by tide fluctuations and fluctuations at high water levels were influenced by the rainfall-runoff component. The rate at which each component influences the water-level time series is approximately 57.40% for the tide component and 27.62% for the rainfall-runoff component, which together account for more than 85% of the influence exerted by all the components together. The wave component is the next most influential at approximately 12.52% and the noise component was found to have very little influence at less than 3%. In order to confirm that the time series of the stream water level are properly decomposed, the time series for the 2012 typhoon 'Sanba' is shown as Figure 10. Tide component is a periodic data showing no variation. Wave component and runoff-rainfall component showed a rising characteristic when rainfall occurs. When the typhoon 'Sanba' occurred, the stream water level was more than 2 m, which indicates that the stream water level time series was properly decomposed. If the Ulsan stream water level station had been situated a little further upstream from its actual location in the estuary, then the influence of the rainfall-runoff component would have been greater than that of the tide component.

**Table 1.** The rate of influence for each component.

| Component | Rate of Influence (%) | Entropy Information |
|---|---|---|
| Tide | 57.40 | 2.42 (33.9%) |
| Wave | 12.52 | 2.39 (33.5%) |
| Rainfall-runoff | 27.62 | 1.43 (20.1%) |
| Noise | 2.46 | 0.89 (12.5%) |
| $\Sigma$ | 100 | 7.13 (100.0%) |

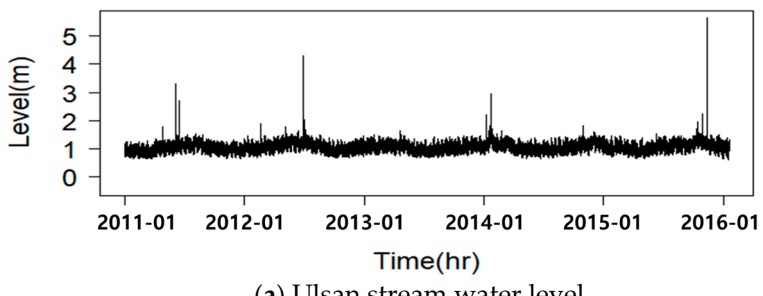

(**a**) Ulsan stream water level

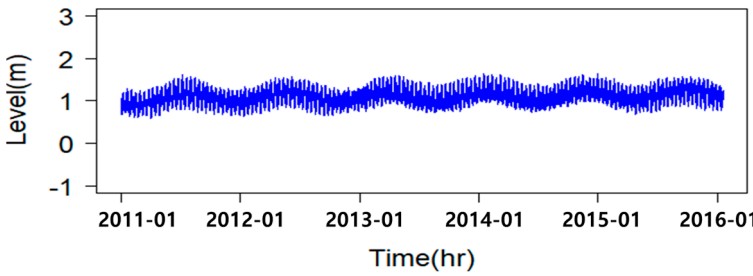

(**b**) Total tide component

**Figure 9.** *Cont.*

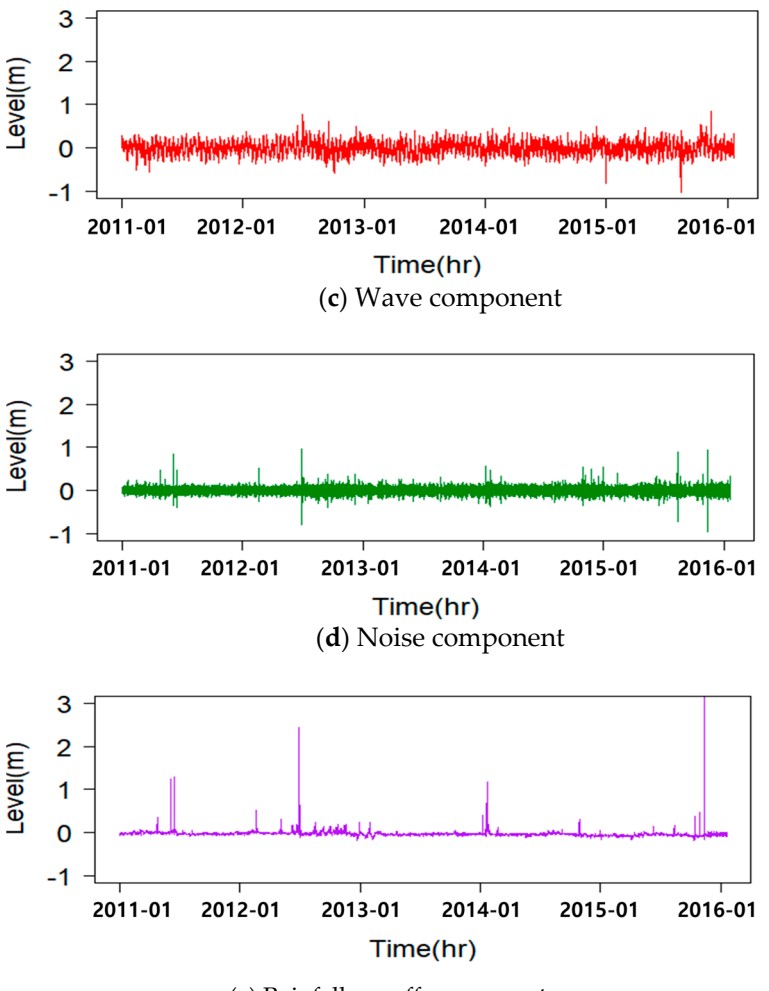

(**c**) Wave component

(**d**) Noise component

(**e**) Rainfall-runoff component

**Figure 9.** Separation results for the stream water level series.

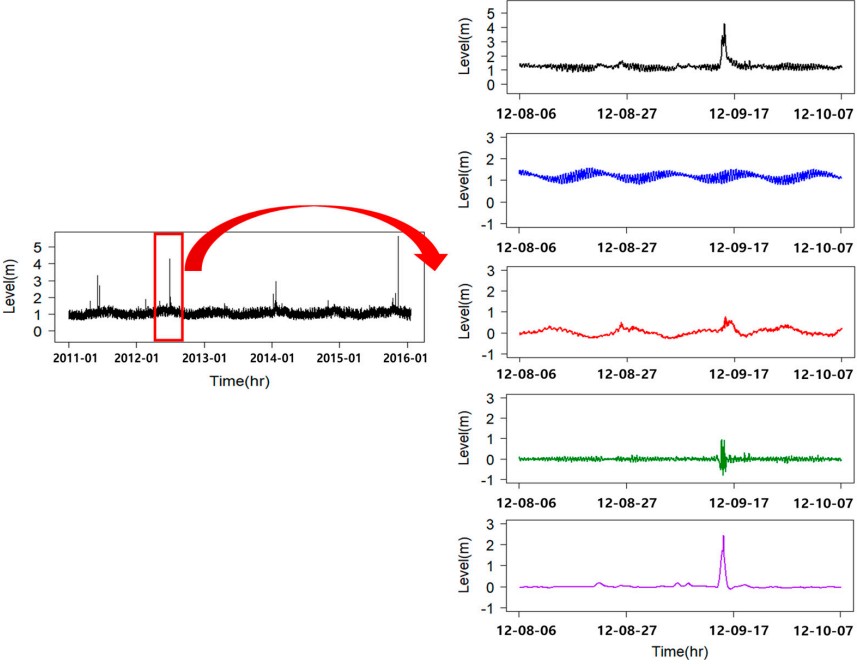

**Figure 10.** Validation of time series decomposition (Typhoon 'Sanba').

Entropy theory was applied to estimate how much information at the stream water level series can be obtained from each component [19]. Markus et al. and Kim et al. used the mutual information theory of entropy to evaluate the stream water level at a station [19,26]. This theory was also used in the present study [26].

The analysis results from the entropy theory for the tide, wave, rainfall-runoff and noise components sending information to the stream water level were about 2.42, 2.39, 1.43 and 0.89 respectively. The results were a little different compared to the rates of influence from the statistical method. The rainfall-runoff component had the second highest influence rate but it was ranked third in terms of the amount of information. The reason is judged to be the characteristics of the data series in each component. The rainfall-runoff component has a large effect on the water-level series when the event occurs occasionally but the amount of information on the stream water level series is small because the uncertainty is large compared to the wave component. For the noise component, which is considered to have the greatest uncertainty, the information on the stream water level series was the smallest at 0.89 (12.5%).

To assess the results, we compared the rainfall-runoff component of the stream water level data from the Ulsan station with the stream water level data from the Guyeong station located upstream, as shown in Figures 11 and 12. The stream water level sensitivity to rain events is different at the Guyeong station given its characteristics but the data patterns are quite similar. Guyeong station is located upstream of the Ulsan station that can be seen from Figure 3, which means that the stream water level of Ulsan station is influenced by the stream water level of Guyeong station. That is, there is a high degree of correlation and similarity when we standardize and compare the two sets of time series data, as shown in Figure 12. The correlation was 0.754, the mean absolute error (MAE) was 0.068 m and the RMSE was 0.114. Thus, the components that influence the stream water level of a tidal river can be decomposed effectively through the proposed methodology.

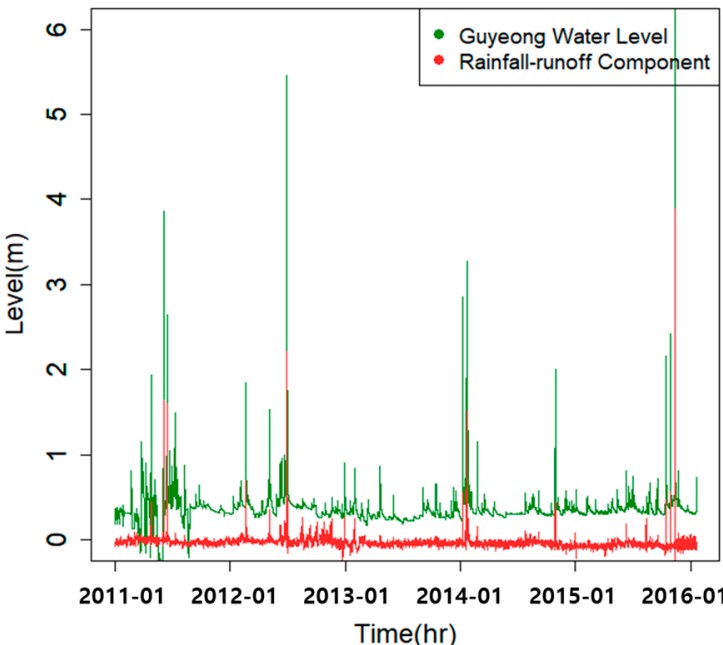

**Figure 11.** Comparison between Guyeong stream water level and rainfall-runoff component.

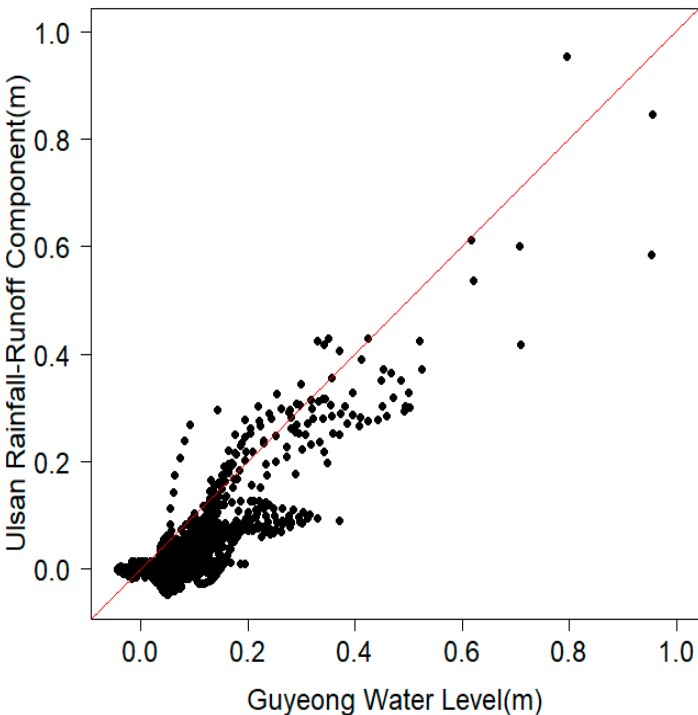

**Figure 12.** Scatter plot of standardized stream water level and rainfall-runoff component.

Nevertheless, the present study has some limitations and room for improvement. First, if a sufficient amount of time series data is obtained, it will be possible to overcome the procedural complexity of decomposing the series into short-, mid- and long-period components and integrating them. Second, the methodology can be applied only if the variability characteristics of the tidal river's water-level time series and sea-level time series are close enough to coincide with one another. Third, Figure 5 shows that the 24-h component showed a significant result but this component could not be extracted because it did not satisfy the 95% significance level. However, since the 24 h component is closely related to the tide, it is necessary to extract this component using another methodology in the future. Finally, if there is a station in an upstream that shows the diurnal cycle, the effect should be considered when this approach is applied.

## 5. Conclusions

This study divided the components that influence the stream water level of a tidal river into tide, wave, rainfall-induced runoff and noise components and proposed a methodology to isolating each one. To this end, we used the data available from hydrological stations in the vicinity of a tidal river and effectively separated the tide and wave components through wavelet analysis. The rainfall-runoff and noise components were separated through filtering analysis. Thus, this study could contribute to the understanding of the behavior of water-level time series in a tidal river, which has a complex structure due to the interaction of various components. Follow-up studies are expected to predict the stream water level of a tidal river while considering the variability characteristics of each component in connection with climate factors. Furthermore, the results of this study and predicted water levels could be used to manage tidal river water levels during the flood season, which could make it possible to minimize the damages due to water disasters such as typhoons.

**Author Contributions:** This research was carried out in collaboration between all authors. S.K. and M.L. had the original idea for the study. K.T.K. provided the research methods and H.S.K. arranged the research. M.L. with Y.Y. conducted the research and drafted the manuscript. All authors discussed the structure of the manuscript and commented on it at all stages.

**Funding:** This work was supported by the Korea Agency for Infrastructure Technology Advancement (KAIA) grant, which is funded by the Ministry of Land, Infrastructure and Transport (Grant 18AWMP-B127555-02).

**Conflicts of Interest:** The authors declare no conflict of interest.

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
