# Peer review of "Decomposition of Water Level Time Series of a Tidal River into Tide, Wave and Rainfall-Runoff Components"

_water, doi:10.3390/w10111568_

Round 1
Reviewer 1 Report
An interesting study, with some potential. Its discourse started clear, but ended unclear.
Observations:
Line 37 “according to the tidal effect [6]. Kim and Kim studied the spatiotemporal variability characteristics” – you should cite the year for “Kim and Kim”.
Line 123 “methodology in this is area.” – remove “is”.
Line 132 “The sea level time series data from the Ulsan sea level station did not show a wide range of fluctuation” – I would rather use the plural of “fluctuation”
Line 134 “The variability of the time series data from the Ulsan water level station was 5.72, that of the Guyeong water level station was 1.99, and that of the Ulsan sea level station was 2.15.” – which is the measure of this variability? I this the standard deviation or other measure? Please specify or define.
Line 137 “data from the Ulsan water level station and sea level station had a correlation coefficient of 0.85” – I suppose it is the Pearson correlation coefficient; please specify it.
146 “Figure 4. Time series of Ulsan and Guyeong stations.” – all figures in this manuscript should be as wide as the space used for main text in order to have more readable fluctuations of the time series (even if many cycles are included).
Line 158 “sea level can be decomposed into tide and wave components. Therefore, the water-level”… Other lines “water level and sea level” AND the title – the authors should replace “water-level” and “water level” with “streamwater level” or “river level” as these are the proper examples for comparison with the “sea level”
Fig.5 b - why aren’t you also using the diurnal tide/cycle for decomposition?! It is clearly present as seen in the scalogram; the diurnal (~24h) fluctuation may be of tidal origin and/or of evapotranspiration origin in the river catchment (the perfect 24h cycle). It is a periodic fluctuation and SHOULD be included in the tidal decomposition (even if you may say it was not observed in pure sea level fluctuations).
Fig.5&6 and other figures – the figure captions should be more self-explanatory
Lines
“199 and the noise component are left (see Fig. 8(a)). We applied a filter function to separate the runoff
200 component, which occurs in response to non-periodic rainfall events, and the noise component,
201 which exhibits random characteristics. We applied a high-pass filter to separate the high-frequency
202 noise component from the relatively low-frequency rainfall-runoff component.”
- quantitative descriptors of the filters should be detailed (I order to have other people able to use the same filters); also, you may wish to continue the pleasant way of describing in detail and step by step these decompositions, as you did for the tidal decomposition.
Lines
“221 The rate at which each component influences the water-level time series is approximately
222 57.40% for the tide component and 27.62% for the rainfall-runoff component, which together account
223 for more than 85% of the influence exerted by all the components together. The wave component is
224 the next most influential at approximately 12.52%, and the noise component was found to have very
225 little influence at less than 3%.”
AND “Table 1. 255 The rate of influence for each component.”
- here or in the methodology section it is advised to describehow exactly had you obtained these percentages; I am pretty sure somebody else may consider another calculation method is better; however, unless you specify how exactly (calculus) you got these numbers, these are useless percentages.
Lines
“235 The analysis results from the entropy theory for the tide, wave, rainfall-runoff, and noise
236 components sending information to the water level were about 2.42, 2.39, 1.43, and 0.89 respectively.”
- the applied entropy theory is not well-known; you must describe the process of obtaining these numbers in the methodology section.
Fig 10&11 – they are not relevant for the proposed demonstration; instead of them and for the entire manuscript, I suggest a systematical comparison of the upstream station with the downstream station.
Overall, for the entire manuscript, I think that selecting (in addition) a case study of a few consecutive months or weeks from your six-years time series will be helpful for indicating, at least graphically, how efficient is your decomposition technique.
Author Response
Thank you very much for your time and insightful review. We have revised attentively the manuscript in order to include your comments.

Reviewer 2 Report
I congratulate the authors for the research carried out, as I find it to be of high quality, and great interest for WATER Journal.
The authors present an interesting research in which tidal river water-level time series are processed in order to evaluate and identify the importance of four different components, i.e., tide, wave, rainfall-runoff and noise. The application presented uses adequate techniques for the purpose of the research.
The study framework is clearly explained and developed, as well as the methods employed.
In my opinion, the research can be very interesting for researches and hydrology engineers.
I have a couple of remarks that might help to improve the manuscript.
Obviously, the whole contribution and the analysis performed is based on water level and sea surface level data from the stations shown in figure 3. Thus, a good initial presentation of such time series is relevant, providing to the reader a first qualitative idea of the characteristics of the time series. Kind of fluctuations, trends, etc. In this respect, figure 4 (a,b,c) could be significantly improved: Size of numbers (too big), time intervals, …. Why not use “days”, weeks, … as 40.000 hours is not very indicative. These comments applied to further figures in the manuscript (7-8-9)
Visualization of original time-series can be significantly improved.
Being so that actually the time resolution is high (not “days” or “weeks”), why not add an additional figure showing in detail some the episodes (with high levels) where the reader could see the actual kind of fluctuations and patterns showed by the time-series with the actual time resolution they have?
Formally speaking, also figures 5 and 6, which are very relevant for the research results presentation, can be improved. Letters, number size, … can be changed for a better visualization. Maybe scale of the graphics.
There is one concern that is not mentioned at all in the manuscript, and maybe deserves some consideration. I wonder whether there is rainfall information in the basin. I guess so. In that case, being clear the rainfall role in water levels from a hydrological point of view, the question is why authors did not used rainfall data in the methodology. Or how useful could it be their inclusion in the analysis for a better differentiation of the corresponding component, maybe using different techniques from the ones presented herein.
Some discussion about this issue might be of interest.
Author Response

(The authors gave the same response as above.)

Round 2
Reviewer 1 Report
Turnitin software has detected that almost the entire section "2.4. Entropy Method for Information Measurement" was copied from a paper in Water and authored by 2 of the authors of this manuscript (partial/self-plagiarism): "Application of the Entropy Method to Select Calibration Sites for Hydrological Modeling".
It is strictly necessary that you modify this section in order to express the same ideas with other words.
Fig.8 - captions, last word: replace "Nosie" with "Noise".
Concerning the diurnal cycle that was detected only as a not-significant signal, this cycle is one of the fewest that we are certain that it exists but our random methods used for detecting significance do sometimes fail to detect it as significant. I insist that at least the Conclusions sections should include a paragraph telling that this cycle is present and that its inclusion in future decomposition techniques might represent an expected improvement of the analysis methods.
This is why I previously requested a more comprehensive comparison between upstream and downstream stations: you included in the rainfall-runoff component not only the surface runoff, but also the baseflow, which is the origin of the diurnal cycle in the whole flow. Using the sea level behaviour to decompose a water level from a tidal river station without taking into account the diurnal cycle, which may be much more visible at the upstream station, will make decomposition less accurate.
I think it is necessary to insert somewhere a graphical comparison of a 7-14 days runoff data of the Ulsan and Guyeong river stations and let readers decide how to interpret your methods. Note: the selected time interval for this new case study should be from "normal" days (baseflow, no floods or rainfalls, preferably with low tides downstream) and when a diurnal cycle is clearly present upstream.
Author Response

(The authors gave the same response as above.)
